# Effects of 12-Month Interdisciplinary Interventions in 8- and 9-Year-Old Children with Excess Body Weight

**DOI:** 10.3390/ijerph192315899

**Published:** 2022-11-29

**Authors:** Dominika Raducha, Joanna Ratajczak, Tomasz Jackowski, Anita Horodnicka-Józwa, Justyna Szmit-Domagalska, Mieczysław Walczak, Elżbieta Petriczko

**Affiliations:** 1Department of Pediatrics, Endocrinology, Diabetology, Metabolic Diseases and Cardiology of Developmental Age, Pomeranian Medical University, 70-204 Szczecin, Poland; 2Institute of Physical Culture and Health Sciences, University of Szczecin, 70-453 Szczecin, Poland

**Keywords:** childhood obesity, lipid profile, nutrition, behavioral intervention, glucose metabolism

## Abstract

Childhood obesity remains one of the most serious medical challenges of the 21st century. The aim of the study was to obtain epidemiological data on the prevalence of overweight and obesity among 8- and 9-year-old children in Szczecin, and to evaluate the effectiveness of medical intervention in the form of a year of interdisciplinary work with children with excess body weight. The study consisted of two main stages: I—screening, II—intervention. The program was implemented for three consecutive years, starting in 2016–2018. The entire population of 8–9-year-olds in Szczecin is 11,494 children. In the screening part of the study, 4890 children took part, whose parent agreed to participate (42.54%). In the intervention part of the study, we analyzed a group of 515 children. Children were further divided into subgroups according to the number of visits completed. Anthropometric parameters were measured on each visit. The prevalence of overweight and obesity in the screened population was 16.9% and 6.4%, respectively. Statistically significant changes were observed in BMI (Body Mass Index) percentiles and BMI z-scores, as well as WHR (Waist-Hip Ratio) during the one year observation time. The best effects were achieved by the 3rd visit (for the first 6 months of the program). Thereafter, the effects diminished due to the longer interval between the 3rd and 4th visits (6 months). There is the need for long-term programs for the prevention of excessive body weight in children and adolescents with frequent checkpoints.

## 1. Introduction

Obesity is a multifactorial systemic disease and a global problem of medicine, public health, and individual concern for a significant percentage of people around the world. Despite the fact that the problem of hunger and underweight is still very serious and unresolved, the majority of the world’s population lives in countries where deaths due to overweight rather than underweight are more common (with the exception of parts of sub-Saharan Africa and Asia). Childhood obesity remains one of the most serious medical challenges of the 21st century, especially due to the possibility of complications in childhood that can affect the quality and length of life [1].

The high percentage of children with excessive body weight is alarming, the scale of this problem indicates an epidemic of obesity among both adults and children. It is necessary to conduct research on obesity in children with particular attention to the realization of prevention and intervention programs. That is why we decided to conduct such a large, population-based study aimed at prevention of excess body weight, so that as many children as possible could achieve real benefits. The literature related to the topic of the effectiveness of prevention programs is abundant, and the results of studies are inconclusive. It is hard to compare programs with each other because of the variety of interventions and the involvement of different patient settings. In addition, limitations in interpretation arise from the lack of data on long-term effects, as well as on the cost-effectiveness of ongoing programs. A review of studies on childhood obesity prophylaxis by Waters et al. that sought to determine the effectiveness of evaluated preventive interventions targeting children with excess body weight appears promising. Effectiveness was assessed based on change in BMI. Thirty-seven studies involving children in the following age groups: 0–5, 6–12, and 13–18 were included in this meta-analysis. This study demonstrated the effectiveness and real impact of prevention programs in children, particularly for programs targeting children aged 6 to 12 years [2]. The results of a large English WAVE study published in 2017, which aimed to assess the effectiveness and efficiency of engaging school environments in supporting families among 6- and 7-year-old children in preventing obesity, are less optimistic. The children participated in a 12-month specialized intervention. However, the authors concluded, the mean BMI score was non-significantly lower compared to the control group at follow-up points 15 and 30 months after the start of the multidisciplinary intervention. The cited analyses suggest that this empirical intervention had no effect on BMI score and thus on childhood obesity prevention [3]. As the child grows older, parental involvement and the effectiveness of family intervention strategies declines, confirming that they are most effective for children under the age of twelve, while better effectiveness of school-based interventions was observed for older children [4].The uniqueness of our project lies in the fact that the screening stage involved a very large population (4890 children) and each child received an individually prepared report on potential abnormalities in the tests conducted. In addition, the project had a very high social impact and was the first such study conducted in our region.

## 2. Aim of the Study

The aim of the study was to obtain data on the prevalence of overweight and obesity among 8- and 9-year-old children in Szczecin, and to evaluate the effectiveness of dietary a behavioral intervention in the form of a year of interdisciplinary work with children with excess body weight.

## 3. Participants and Methods 

### 3.1. Study Design 

The study was prospective in nature. The study group included children participating in a program aiming to decrease excess body weight in Szczecin children, called “the Brave Eight”. The program was implemented by Independent Public Clinical Hospital No. 1 of the Pomeranian Medical University in Szczecin, the University of Szczecin and SZKOLMED, a Non-public Health Care Institution of School Medicine in Szczecin. The study consisted of two main stages: I—screening, II—intervention. The program was implemented for three consecutive years, starting in 2016 to 2017 and 2018. In 2016 we screened children born in 2008, in 2017 those born in both 2008 and 2009, and in 2018, those born in 2010. The program was fully financed by the Municipality of Szczecin. It was created under license from the Polish Society for Health Programs in Gdansk who evaluated our work positively. The Society was also the auditor of the project. The conduct of the study was reported to the Bioethics Committee at the Pomeranian Medical University in Szczecin, receiving a positive opinion no. KB-0012/85/15 dated 22 June 2015.

#### 3.1.1. I Stage of the Study (Screening)

The first stage of the program was made up of screening and comprehensive health analysis. All eight- and nine-year-old children in Szczecin (attending second and third grades) whose parents agreed to participate in the program were examined. The examinations took place in every elementary school in Szczecin: this included, public, private and special schools. They were conducted by a qualified, trained team of school nurses who continuously entered patient data and their test results into a computer program prepared for this project. The examinations during the first stage included measurements of basic anthropometric parameters, such as: body height, body weight, waist and hip circumference and also BMI analysis. The following were also assessed: blood pressure, body composition analysis and cardiorespiratory fitness (physical fitness). BMI and WHR were calculated based on the measurements obtained. 

#### 3.1.2. Selection Criteria 


Inclusion criteria


Children with excessive body weight who were diagnosed with a BMI ≥ 90 percentile qualified for the next stage of the program. The results were related to Polish percentile grids according to the OLAF and OLA study conducted in 2007–2012 [5]. Criteria developed by the CDC (Centers for Disease Control and Prevention) were adopted were adopted: overweight: BMI ≥ 85–95 c, obese: BMI ≥ 95 c [6]. 


Exclusion criteria


Participants who reduced their BMI percentile < 90 according to the percentile charts from the OLA and OLAF [5] kończyli udział w programie. Another exclusion criterion was age < 6 and >10 years. 

#### 3.1.3. II Stage of the Program (Intervention)

Children who qualified for the intervention section remained under one year of interdisciplinary care. During this period, they attended four specialist visits. The data was divided into subgroups: medical data concerning children on the first visit—group 1; medical data concerning children on the second visit—group 2; medical data concerning children on the third visit—group 3; medical data concerning children on the fourth visit—group 4. The planned interval between the 1st and 2nd visit was 3 months and the period between the 2nd and 3rd visit was also 3 months, while the interval between the 3rd and 4th was longer, at 6 months. During each meeting, the child had four visits with the following specialists: a doctor, a nutritionist, a physical activity specialist and a psychologist. By design, during the one-year program, each child would have 16 specialist consultations, to which they attended with a minimum of one caregiver. Whole families were invited, especially those who were directly responsible for the child’s nutrition process. Anthropometric parameters, blood pressure, body composition analysis and cardiorespiratory fitness (physical fitness) were reassessed. Furthermore, in order to promote an active lifestyle and show alternative ways of spending leisure time, various sports activities were organized: at a swimming pool, a multi-purpose rally in the woods, soccer training with players from Pogon, the Szczecin football club, as well as general development activities in the gym and a family bicycle trip. Knowing that the foundation for changing lifestyles is the support of parents and guardians, we prepared several hours of educational workshops for this group, conducted by a doctor, a nutritionist, a psychologist and a physical activity specialist. A series of educational training sessions were also conducted for teachers and school cafeteria workers. 

### 3.2. Study Population

#### Projected and Real Number of Children 

The number of children of the years expected to be examined was 11,494 [7], a figure we obtained from the Szczecin City Hall Ultimately, 4890 Szczecin children were examined during the program—that is why many parents agreed to participate in our project. Eight hundred and twenty-six children were diagnosed as overweight, which accounted for 16.9% of all children examined in schools, while three hundred and thirteen children were diagnosed as obese, which accounted for 6.4%. A total of 23.3% of the children screened were found to be overweight. Due to the social nature of the program, older or younger children who attended the class where the survey was taken, were also examined, so that no child felt left out. For this reason, the study group includes several children younger and older than expected. The statistical analysis included data mainly from 7-, 8- and 9-year-old children and also a total of several 6- and 10-year-olds. For the second stage of the program, 745 children with a BMI ≥ 90th percentile qualified, accounting for 15% of the studied population, including 408 girls and 337 boys. After excluding children who scheduled the first appointment and did not attend and those >10 years old, the group was left with 573 children who attended the first visit. We also excluded those children from this group who, between the first and second stages of the program, reduced their BMI centile to such a degree that they no longer met the eligibility criterion for the second stage, i.e., BMI ≥ 90c (58 children). In the end, during the 3-year course of the study, we analysed a group of 515 children, including 1 child aged 6 years, 148 children aged 7 years, 233 children aged 8 years, 127 children aged 9 years and 6 children aged 10 years—(Figure 1).

### 3.3. Methods

#### 3.3.1. Anthropometric Parameters

Anthropometric parameters were assessed in the same way during the first screening stage and at each of the four visits during the intervention stage. Body height was assessed using a Harpenden-type stadiometer with an accuracy of 0.1 cm. During the measurements, children were placed in the Frankfurt position. Body weight was assessed using a body composition analyzer with an accuracy of 0.1 kg. Waist and hip circumference were assessed using a sewing tape measure with an accuracy of 0.1 cm. 

#### 3.3.2. Blood Pressure

Arterial pressure was measured three times using an Omron 2 electronic blood pressure monitor. Measurements were taken after several minutes of rest in a sitting position, several minutes apart. It was also important to reassure the child and alleviate any anxiety to avoid false readings. Before the measurements were taken, the children sat in a quiet examination room on a chair with a backrest. Both feet of the participants lay flat on the floor with the legs uncrossed, and they maintained this sitting position for at least 5 min. Thirty minutes before the measurements, the patient did not exercise. The children’s blood pressure was assessed in the same way during the first screening stage and at each of the four visits during the intervention stage. Next, an arm cuff was selected and placed directly on the patient’s arm, with no clothing under the cuff and no clothing sleeves rolled up above the cuff. Once the cuff was positioned, the patient’s arm was supported so that the center of the cuff was at the level of the right atrium. The patient was not spoken to during the measurements [8].

#### 3.3.3. Body Composition Analysis

During the screening stage, body composition analysis was performed with a Jawon Medical X- Contact 350 body composition analyzer, while during the intervention stage, Jawon Medical IOI 353 was performed using the electrical bioimpedance (BIA) method. The following parameters were determined: percentage of body fat, fat mass, lean body mass, water mass and muscle mass.

#### 3.3.4. Physical Fitness

The Kasch Pulse Recovery Test Step, a test that determines post-exercise heart rate, was used to assess cardiorespiratory fitness. The test consisted of rhythmically ascending a 30.5 cm platform for 3 min. The pace was 24 ascents and descents determined by a metronome. The heart rate was recorded using an electronic analyzer from “Polar” and monitored throughout the test, i.e., 3 min of exercise load (step-test) and 1 min and 5 s during rest (sitting position). Only the values recorded for one minute, immediately after the test (but no later than 5 s after the cessation of exercise) were analyzed. During rest in the sitting position, all indications of the heart rate monitor in the tested child were recorded. The arithmetic mean calculated from these figures was the primary variable of the analyses made. Before the start of the step-test, ascents and descents at an appropriate pace were demonstrated by the nurses or physical activity specialists conducting the study. The stopwatch was turned on only after 3–4 attempted steps by the patient. The study provided an estimate of the level of physical fitness based on HR (heart rate) frequency, and thus, the level of aerobic physical activity, which is an essential component of a healthy lifestyle.

#### 3.3.5. Doctor’s Office

In the doctor’s office, a detailed medical history was taken, in addition to which the patient was also subjected to a physical examination, with a special focus on examination deviations. Again, anthropometric parameters were assessed, as well as blood pressure measurement, body composition analysis and physical fitness assessment. At each visit, the patient was given individual recommendations. 

#### 3.3.6. Physical Activity Room 

Patients performed physical fitness and physical capacity tests during visits to the physical activity room. During the tests, abdominal muscle strength, shoulder girdle muscle strength, flexibility, long jump from a standing position, as well as physical fitness were examined. In addition, the physical activity specialist conducted an interview, during which they assessed answers to questions related to the child’s participation in physical education classes, time spent in front of the TV and computer, and how they commute to and fro school. Based on the results of the test, the physical activity specialist provided the patient and their caregiver with recommendations that addressed improvements in muscle corset, motor coordination or endurance. They also addressed improvements in quality of life related to active recreation, spending leisure time outdoors or encouraging participation in organized group sports.

#### 3.3.7. Dietitian’s Office

Consultations with the dietitian primarily included nutritional education, and included individualized recommendations for the patient. A dietary interview created specifically for the program was conducted by the dietitian during each visit. It included an assessment of: frequency of consumption and quality of breakfast, amount of water drunk, consumption of sweets and sugary drinks, vegetables, fruits, dairy products, fish, nuts and seeds, pulses, fast food, processed foods, cooking techniques used, and regularity of meals. Based on the collected history, specific dietary recommendations were selected. During each visit, the dietitian monitored the course of changes in lifestyle behavior and focused on a different aspect of healthy eating so that changes could be made gradually.

#### 3.3.8. Psychologist’s Office

Psychological consultation was based on a thorough understanding of the child, his family situation, his functioning in society, forms of coping with stressful situations, eating habits of the whole family. Together with the child, parents and specialist, new rules related to nutrition were established. In addition, patients and their families were supported and motivated in following the recommendations, achieving the set goals, and spending their leisure time actively. Emphasizing the point that the change should involve the whole family and not just the child.

## 4. Statistical Analysis Methods

Stata 11 (license number 30110532736) and MS Office Excel were used in the statistical analysis. Continuous variables were checked for distribution normality using the Kolmogorov–Smirnov test. The variables were described as mean values, deviations and standard errors, as well as minimum and maximum values. The variables were described as mean values, deviations and standard errors, as well as minimum and maximum values. The results were described by the correlation coefficient r and the probability p. In all tests performed, statistically significant differences were considered those for which the probability—p was equal to or less than 0.05. A significance level of *p* = 0.051–0.099 was designated as a trend bordering statistical significance.

## 5. Results

### 5.1. Basic Anthropometric Parameters and Their Variables in the Study Group at the Time of Four Specialized Visits

The study group consisted of 515 children with excess body weight, including 278 girls (53.9%) and 237 boys (46.1%). Overweight was diagnosed in 238 children, accounting for 46.2%, and obesity in 277, accounting for 53.8%. In accordance with the objectives of the preventive program for children, four visits were scheduled. Of the entire study group, 515 children only attended the first visit (group 1), 394 children attended the first and second visits (group 2), 278 children attended the first, second and third visits (group 3) and 195 children attended all four specialized visits (group 4). During all visits, auxological parameters of the study population were assessed. Detailed auxological data of the study group are provided in Table 1.

Changes in the auxological parameters of the study population were also evaluated during all visits. Parameter changes were calculated in such a way that for each parameter, the difference between each was initially calculated for specific participants, and then the average was calculated from the data obtained. Differences for all parameters were calculated in the same way. Detailed auxological data are provided in Table 2.

### 5.2. Time Variables

The time measured in days elapsing from the first visit to the second visit was determined in 394 children, which accounted for 76.5% of the total study population. The interval between the 1st and 3rd visit was marked in 278 children, which accounted for 54% of the study population, while the interval between the 1st and 4th visit was marked in 195 children, which accounted for 37.9%. It is noteworthy that with each successive visit the interval increased and the attendance decreased. Detailed data are provided in Table 3.

### 5.3. Correlations between Anthropometric Parameters in Children in the Second Group

The analyzed group included children who had their first and second specialist visits and ended their participation in the program at this stage. In the analyzed group, statistical significance was established for BMI centile (*p* < 0.01, R = 0.18) and BMI z-score (*p* < 0.01, R = 0.15). It is noteworthy that both BMI centile and z-score BMI decreased at the 2nd visit. No statistical significance was observed for waist and hip circumference and WHR in the study group. Detailed data are provided in Table 4.

### 5.4. Correlations between Anthropometric Parameters and Their Variables in Children in the Third Group

The analyzed group included children who had their first, second and third specialist visits and ended their participation in the program at this stage. As in group two, statistical significance was established for BMI percentile (*p* < 0.01, R = 0.20) and BMI z-score (*p* < 0.01, R = 0.17). A decrease in BMI centile and BMI z-score was noted. Statistical significance of BMI centile was observed between the 1st and 2nd Visit (*p* < 0.01), 1st and 3rd Visit (*p* < 0.01). Alternatively, the correlation between the 2nd and 3rd visits was at the limit of statistical significance. Statistical significance of z-score BMI was also observed between the 1st and 2nd visits (*p* < 0.01) and 1st and 3rd visits (*p* < 0.01), while it was not observed between the 2nd and 3rd visits. Of note, a lower BMI centile and z-score BMI were still seen at the third visit, than at the first visit. Initially, a decrease in hip circumference was noted, while a statistically significant increase was observed at the third visit (*p* = 0.05, R = 0.09). Statistical significance in hip circumference was observed only between the 2nd and 3rd visits (*p* = 0.02). No statistical significance was observed for waist circumference in the study group, while WHR between the 1st and 3rd visits was close to statistical significance. Detailed data are provided in Table 5.

The change in anthropometric parameters in children in the third group was also analyzed. For individual anthropometric parameters, the difference between them was initially calculated for specific participants between the 1st and 3rd visits, and then the mean was calculated from the data obtained. Statistical significance was established for BMI percentile change (*p* < 0.01, R = 0.82), BMI z-score change (*p* = 0.01, R = 0.11), waist circumference change (*p* = 0.04 R = 0.09) and hip circumference change (*p* < 0.01, R = 0.17). The change in WHR was not statistically significant. Detailed data are provided in Table 6.

### 5.5. Correlations between Anthropometric Parameters and Their Variables in Children in Group Four

The analyzed group included children who had all specialized visits, i.e., completed the program. As in group two, statistical significance was established for BMI percentile (*p* < 0.01, R = 0.20) and BMI z-score (*p* < 0.01, R = 0.17). Additionally, there was a decrease in BMI centile and BMI z-score. BMI remained on the border of statistical significance in this group. In contrast, BMI values between visits 2 and 4 were statistically different *p* = 0.01. A decrease in BMI percentile was observed from one visit to the next. The highest mean BMI percentile was observed at the first visit and averaged 95.4 ± 2.4, while the lowest was observed at the fourth visit and averaged 93.4 ± 4.6. These data were statistically significantly different from each other (*p* < 0.01, R = 0.21). Statistical significance was observed between the 1st and 2nd visits (*p* < 0.01), 1st and 3rd visits (*p* < 0.01), 1st and 4th visits (*p* < 0.01) and 2nd and 4th visits (*p* < 0.01), while the relationship between the 2nd and 3rd visits was on the border of statistical significance. No statistical significance was established between visit 3 and visit 4. Similarly, the z-score BMI also presented a decrease in values from one visit to the next. The highest mean z-score BMI was observed at the first visit and averaged 1.9 ± 0.3, while the lowest was observed at the fourth visit and averaged 1.7 ± 0.4. These data were statistically significantly different from each other (*p* < 0.01, R = 0.19). Statistical significance was observed between the 1st and 2nd visits (*p* < 0.01), 1st and 3rd (*p* < 0.01), 1st and 4th (*p* < 0.01), 2nd and 4th (*p* = 0.02). Statistical significance was not established between the 2nd and 4th visits and the 3rd and 4th visits. There was an observed increase in waist circumference in children between the 2nd and 4th and 3rd and 4th visits, which was statistically significant at *p* = 0.02 and *p* = 0.04, respectively. In contrast, the observed decrease in waist circumference between the 1st and 2nd, 1st and 3rd and the increase between the 1st and 4th and 2nd and 3rd visits was not statistically significant. An initial decrease in hip circumference and subsequent gradual increase in hip circumference was demonstrated (*p* < 0.01, R = 0.16). Statistical significance was observed between individual visits 1 and 4 (*p* < 0.01), 2 and 3 (*p* = 0.05), 2 and 4 (*p* < 0.01), 3 and 4 (*p* = 0.03). Analyzing the WHR values at each visit, there was a slight decrease in WHR values with borderline statistical significance, while statistical significance was found for a decrease in WHR values between visits 1 and 3 (*p* = 0.03) and borderline for visits 1 and 4. Detailed data showing the anthropometric measurements in the 4th group are shown in Table 7.

As with the previous groups, the variables of the various parameters were also evaluated in the 4th group. It is noteworthy that the interval between visits gradually increased and was most pronounced in the final part of the program duration, i.e., between the 3rd and 4th visits. The values were statistically significantly different (*p* < 0.01, R = 0.93). In addition, statistical significance was observed between the 2nd and 3rd visit (*p* < 0.01), 2nd and 4th visit (*p* < 0.01), 3rd and 4th visit (*p* < 0.01). The change in BMI decreased the most between the 1st and 2nd visits but also between the 1st and 3rd visits and increased between the 3rd and 4th visits. These values were statistically significantly different from each other (*p* < 0.01, R = 0.23). In addition, statistical significance was observed between the 2nd and 4th visits (*p* < 0.01), 3rd and 4th visits (*p* < 0.01). The change in BMI percentile decreased between the 1st and 2nd visit, the 1st and 3rd visit, and the highest between the 1st and 4th visit. These values were statistically significantly different from each other (*p* < 0.01, R = 0.16). In addition, statistical significance was observed between the 2nd and 3rd visit (*p* = 0.02), the 2nd and 4th visit (*p* < 0.01), while the change in BMI percentile between the 3rd and 4th visit was on the verge of statistical significance (*p* = 0.07). The change in the z-score of BMI decreased between the 1st and 2nd visits, the 1st and 3rd, and the greatest between the 1st and 4th. These values were statistically significantly different from each other (*p* < 0.01, R = 0.17). In addition, statistical significance was observed between visits 2 and 3 (*p* = 0.02), 2 and 4 (*p* < 0.01), while the change in z-score of BMI between 3 and 4 was at the limit of statistical significance (*p* = 0.07). The change in waist circumference decreased the most between the 1st and 2nd visits. There was also a decrease between the 1st and 3rd visits and an increase between visits 1 and 4. These values were statistically significantly different from each other (*p* < 0.01, R = 0.19). Furthermore, statistical significance was also observed between visit 2 and 4 (*p* < 0.01) and between visit 3 and 4 (*p* < 0.01). The change in hip circumference decreased between the 1st and 2nd visits, then an increase was observed between the 1st and 3rd visits and between the 1st and 4th visits. These values were statistically significantly different from each other (*p* < 0.01, R = 0.27). Besides, statistical significance was also observed between the 2nd and 3rd visits (*p* < 0.01), between the 2nd and 4th visits (*p* < 0.01) and between the 3rd and 4th visits (*p* < 0.01). A slight reduction in the change of WHR index was observed but without statistical significance. Detailed data are provided in Table 8 and Figure 2, Figure 3, Figure 4 and Figure 5.

## 6. Discussion

Out of a large, representative population of 4890 mainly 8- and 9-year-old children surveyed, excess body weight accounted for as much as 23%, meaning that according to this study, one in five children in Poland faces an excess weight problem. In this group, overweight accounted for 16.9%, while obesity accounted for 6.4%. The epidemiological situation in our western neighbors appears similar. Two large German surveys, KIGGS in 2007 and CrescNet in 2010, which analyzed the prevalence of excess body weight among children between the ages of 3 and 17, showed overweight in about 15–17% of the subjects, while obesity in about 6.2–7.6%, for a total of 21.1–24.6% [9,10]. Another phase of the German KIGGS survey in 2018, showed a trend of overweight stabilizing but still at a high level. Overweight was found in 15.4% and obesity in 5.9% [11].

Based on the data in Table 2 and Table 3 showing the absolute values of anthropometric parameters in each group and their variables, it can be concluded that the intended effects of improving and inhibiting the growth of individual body mass indices, especially the BMI percentile and BMI z-score, were achieved. In both girls and boys, an increase in body weight was observed during subsequent visits, which is related to the physiological process of child growth. Therefore, special attention in the analysis of anthropometric parameters should be paid to changes in BMI, and especially BMI centile and BMI z-score, which are objective indicators. Based on the analysis of the previously mentioned tables, it is concluded that in the group of all children who participated in the program there is a successive increase in body weight of about 5 kg between the 1st and 4th visits. BMI values decreased by the time of the 3rd visit, with the greatest reduction between the 1st and 2nd visits, while an increase in the index was observed at the 4th visit compared to the 1st visit. Importantly, it is noteworthy that there was a gradual decrease in both BMI percentile and BMI z-score during all visits among all children. Similar results of successive reductions in body mass indices were obtained in a study by Robertson et al. on the basis of an intervention program conducted among German 7- to 13-year-old children 3 and 9 months after the start of the measurements [12]. It should not be forgotten that parameters that may be relevant to metabolic syndrome diagnosis, such as waist circumference and hip circumference, also initially changed favorably between groups, but worsened over time. In the case of waist circumference change, a reduction was observed up to the 3rd visit, with the greatest between the 1st and 2nd visits, and unfortunately an increase in circumference at the 4th visit. For change in hip circumference, a reduction was observed only up to the 2nd visit, with the largest increase in circumference at the 4th visit. The WHR index decreased slightly until the 2nd visit, remaining stable thereafter. There was no statistical significance found for the change in BMI, for the change in BMI z-score between the 1st and 4th visits, for waist circumference change between the 1st and 3rd visits, and for hip circumference between the 1st and 4th visits. Changes in anthropometric parameters in each group from the first visit were analyzed in detail. A statistically significant decrease in BMI percentile and BMI z-score was observed in group 2. A decrease in BMI and a decrease in waist and hip circumference were also observed, but without statistical significance. In Group 3, a decrease in BMI centile and z-score BMI between visits was also observed. In addition, there was a decrease in hip circumference between the 1st and 2nd visits, and a statistically significant increase and decrease in waist circumference between the 2nd and 3rd visits, but no statistical significance in WHR. Most interesting are the changes in anthropometric parameters in group 4, the group of children who completed the entire program, to assess the effectiveness and effects of year-long interdisciplinary cooperation and care. A decrease in BMI percentile and z-score BMI was also observed in group 4. Waist and hip circumference decreased at the initial visits while it increased for hip circumference at the 3rd and 4th visits comparing to the 1st, and it increased for waist circumference at the 4th visit. A decrease in WHR was also noted but without statistical significance. The above data indicate that a decrease in BMI percentile and BMI z-score from the first visit was observed in all groups, as well as a decrease in their variables. There was also an initial decrease in waist and hip circumference, while at the end of the program an increase in waist and hip circumference was observed in each group at different stages depending on the group analyzed. It was also observed that WHR initially decreased slightly with stabilization of values, while it was not statistically significant in any of the groups. It is worth noting that WHR decreased little and remained a prognostic marker of metabolic syndrome with all its consequences. The data I have presented also demonstrate the effectiveness of non-pharmacological treatment. The 2017 WHO international study indicates that among Polish 8-year-old children, values ≥ 90c of waist circumference were reported in 21.5%, observed significantly more often in girls, and hip circumference in 18%, observed more often in boys, but the difference was not statistically significant [13]. In contrast, in this study, a significantly larger circumference of both waist and hips was observed in the boys’ group. It is worth noting that an additional aggravating factor for the patient is the type of obesity. A multicenter study by Yusuf et al. published in the Lancet in 2005 described that WHR in adults showed the strongest association with myocardial infarction and was its strongest anthropometric predictor regardless of sex, ethnicity, smoking, lipid and carbohydrate disorders and carbohydrate metabolism, or hypertension [14].

We can look for an explanation for the question of why, despite the reduction in BMI percentile and BMI z-score, we observed an increase in both waist and hip circumference in the second half of the program. This may be due to the fact that during the course of our activities, children gradually reached the 2nd phase of sexual maturation, which may be associated with a change in body proportions, especially in girls, who mature faster. The most intense improvement and the greatest changes in parameters were seen at the beginning of the program, especially between the 1st and 2nd visits. The success observed initially with a tendency to slow down the improvement or worsening of the parameters could probably be due to the natural behavior inherent in the human psyche, especially that of a child. At the beginning of lifestyle modification, the effects are the best, which is probably due to the greatest commitment, desire and need for change. Over time, it becomes increasingly difficult for patients to achieve the desired effect or to follow the recommendations set. With the passage of time, patients with overweight, who are difficult and demanding patients, become tired and weary of having to follow recommendations. They lack energy and motivation, and as a result, the greatest effects are observed at the beginning of therapy. This is a trend found in many aspects of medicine. Another study by Reinehr et al. shows a reduction in BMI and SDS BMI in 10-year-old children as a result of participation in the Obeldisks intervention program, which aimed to improve body mass indexes. The children significantly reduced parameters at the end of the program and 3 years after its completion. The authors also emphasize that weight reduction in the first 3 months of the program was highly predictive of long-term success [15]. The data presented here confirm our initial goals and objectives of improving body mass indices, stunting and stabilizing them during the one-year intervention and the cooperation of the children and their families, thus confirming the effectiveness of our interventions. For children with overweight or less advanced obesity, the goal to be achieved, and thus the form of success, was to maintain body weight at the same level, with a decrease in BMI or its stabilization achieved with increasing body height and over time. In contrast, weight reduction was recommended for children with advanced obesity, especially those with complications or comorbidities. Based on the results of the previously presented studies, I believe that the population of children that was evaluated has a good chance of achieving long-term success in the form of a decrease or maintenance of previous body mass indexes.

From what has been discussed so far, it appears that the treatment of obesity is only moderately effective, the priority area for future research should be the prevention of obesity in early childhood and the formation of health-seeking behaviors at every stage of a child’s development. Actions to increase knowledge of lifestyle diseases in the local community, may be important in the context of a broader, systemic approach to preventing excess body weight in children and changing wider health policies. Further research is needed to confirm the effectiveness of implemented interventions. Holistic interventions should be sustained and efforts made to prevent obesity in children and adolescents caring for their quality and length of life.

## 7. Study Limitation

The size of the study group however large, 515 children-represented 69% of the children who could potentially benefit from the one-year intervention program. Unfortunately, similar consent percentages were also reported by authors from other clinical centers. Public awareness of what a serious health problem overweight and obesity is, especially in children, is consistently too low.

Recent studies also point to the value of the WHtR (waist to height ratio) index in predicting cardiovascular complications. The authors of the present study used the WHR index. Particularly in the pediatric population, WHtR could more objectively indicate changes in adipose tissue distribution with a particular focus on visceral fat.

## 8. Conclusions

### 8.1. Results Associated with the Prevalence of Excess Body Weight

1. In a representative group of mainly 8- and 9-year-olds screened from the city of Szczecin, the prevalence of excessive body weight was more than 23.3%, of which overweight accounted for 16.9% and obesity for 6.4%.

### 8.2. Results Associated with the Effectiveness of the Intervention

The one-year multispecialty intervention program turned out to be effective, as it led to a halt in weight gain and resulted in improvements in anthropometric indicators such as BMI, BMI centile and BMI z-score mainly in the population of 8- and 9-year-olds with excessive body weight.The best effects were achieved by the 3rd visit (for the first 6 months of the program). During the course of the program the effects diminished in comparison to 1st or 3rd visit, but were still noticeable. It was most likely caused by the longer interval between the 3rd and 4th visits (6 months). There is the need for long-term programs for the prevention of excessive body weight in children and adolescents with frequent checkpoints.

## Figures and Tables

**Figure 1 ijerph-19-15899-f001:**
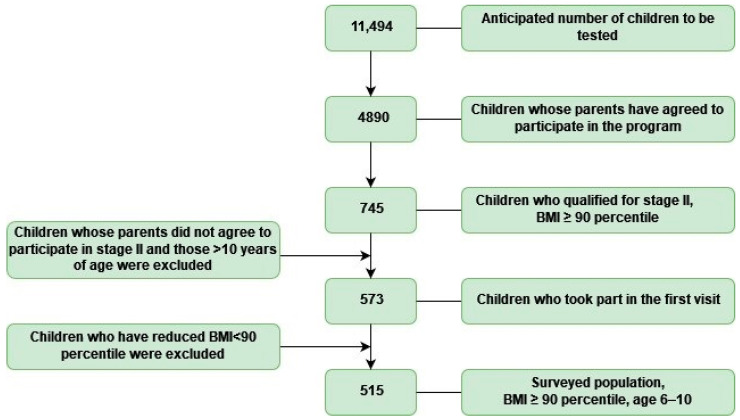
Diagram showing how children qualified for the study group.

**Figure 2 ijerph-19-15899-f002:**
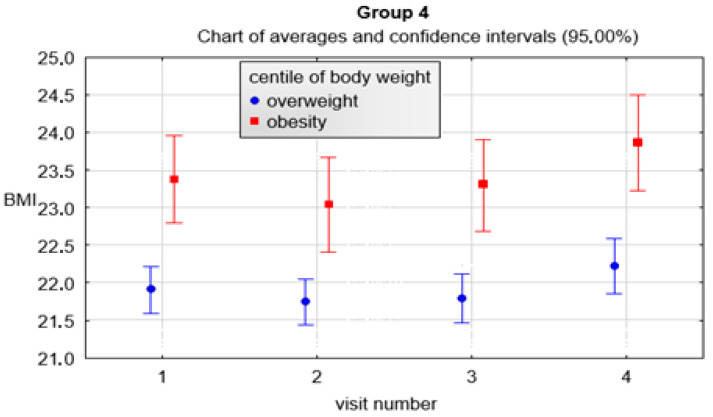
BMI in children with overweight and obesity at consecutive visits.

**Figure 3 ijerph-19-15899-f003:**
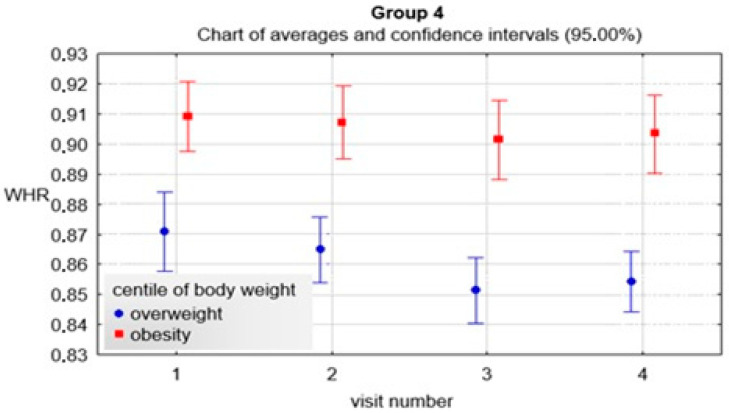
WHR index in children with overweight and obesity at consecutive visits.

**Figure 4 ijerph-19-15899-f004:**
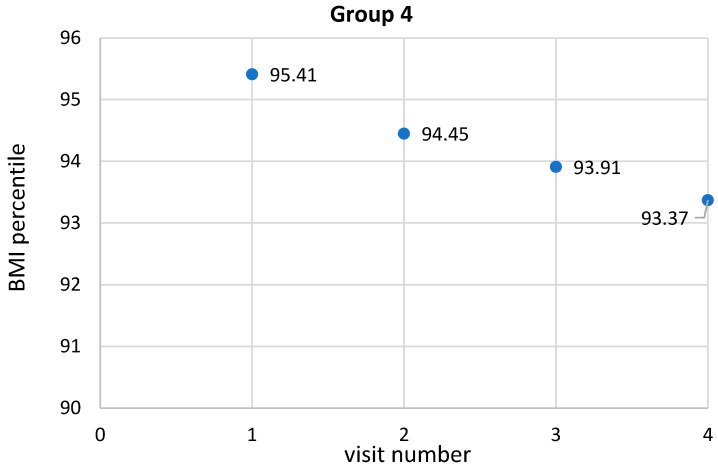
Change in BMI percentile during subsequent intervention visits.

**Figure 5 ijerph-19-15899-f005:**
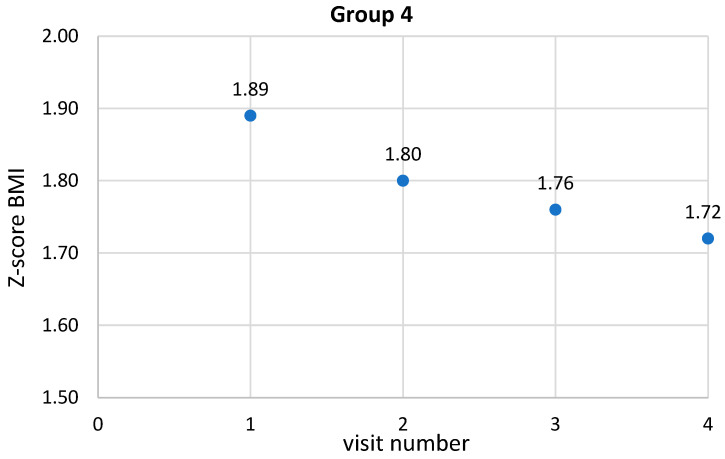
Change in BMI z-score during subsequent intervention visits.

**Table 1 ijerph-19-15899-t001:** Auxological data of the study group.

Parameter	Visit Number	n	$\bar{x}$	SD	Min.	Max.
Age [years]	1	515	8.45	0.71	6.91	10.53
Age [years]	2	394	8.68	0.73	7.16	10.79
Age [years]	3	278	8.98	0.73	7.42	11.19
Age [years]	4	195	9.52	0.73	7.88	11.60
Body height [cm]	1	515	137.01	7.04	117.00	159.60
Body height [cm]	2	392	138.60	7.13	117.80	162.00
Body height [cm]	3	278	140.67	7.17	118.50	163.50
Body height [cm]	4	195	143.75	7.17	121.50	163.50
Z-score of body height	1	515	1.08	0.86	−1.97	3.28
Z-score of body height	2	392	1.11	0.84	−1.98	3.31
Z-score of body height	3	278	1.16	0.83	−2.15	3.00
Z-score of body height	4	195	1.16	0.83	−1.86	3.54
Body weight [kg]	1	515	42.81	7.38	25.60	78.00
Body weight [kg]	2	391	43.21	7.63	26.70	81.80
Body weight [kg]	3	277	44.92	8.12	26.40	87.20
Body weight [kg]	4	195	47.72	8.38	27.20	93.70
Z-score of body weight	1	515	1.99	0.47	0.48	3.24
Z-score of body weight	2	391	1.90	0.48	0.46	3.14
Z-score of body weight	3	277	1.88	0.49	0.35	3.18
Z-score of body weight	4	195	1.83	0.50	0.18	3.24
BMI	1	515	22.65	2.36	18.70	33.80
BMI	2	391	22.34	2.45	17.64	34.85
BMI	3	277	22.53	2.58	18.44	35.60
BMI	4	195	22.95	2.58	18.01	37.34
Z-score BMI	1	515	1.90	0.30	1.33	2.76
Z-score BMI	2	391	1.80	0.32	0.94	2.71
Z-score BMI	3	277	1.76	0.34	0.92	2.70
Z-score BMI	4	195	1.72	0.35	0.67	2.65
Waist circumference [cm]	1	510	72.35	6.84	58.00	96.50
Waist circumference [cm]	2	384	71.55	6.83	54.00	95.00
Waist circumference [cm]	3	276	72.02	7.07	57.00	98.00
Waist circumference [cm]	4	193	73.04	7.14	56.00	101.00
Hip circumference [cm]	1	510	81.30	6.08	64.00	103.00
Hip circumference [cm]	2	384	80.82	5.99	66.00	104.00
Hip circumference [cm]	3	276	82.23	6.36	62.00	108.00
Hip circumference [cm]	4	193	83.42	6.45	69.00	110.00
WHR	1	510	0.89	0.06	0.71	1.16
WHR	2	384	0.89	0.06	0.60	1.08
WHR	3	276	0.88	0.07	0.67	1.45
WHR	4	193	0.88	0.06	0.73	1.28

$\bar{x}$: average; n: size of the group; SD: standard deviation; min.: minimum value; max.: maximum value.

**Table 2 ijerph-19-15899-t002:** Auxological variables of the study group.

Parameter	Visit Number	n	$\bar{x}$	SD	Min.	Max.	*p*
∆ Body height	2	391	0.32	1.86	−17.10	4.50	<0.05
∆ Body height	3	277	1.96	2.94	−14.00	16.00	<0.01
∆ Body height	4	195	5.14	3.59	−11.10	20.50	<0.05
∆ Z-score of body height	2	391	−0.09	0.14	−1.38	0.23	<0.05
∆ Z-score of body height	3	277	−0.12	0.21	−1.20	1.04	<0.05
∆ Z-score of body height	4	195	−0.14	0.22	−1.16	0.84	>0.20
∆ BMI	2	391	−0.27	0.87	−7.46	1.74	<0.15
∆ BMI	3	277	−0.08	1.21	−5.96	3.70	<0.10
∆ BMI	4	195	0.39	1.38	−5.58	4.53	<0.20
∆ z-score BMI	2	391	−0.09	0.14	−1.16	0.25	<0.05
∆ z-score BMI	3	277	−0.13	0.19	−0.87	0.47	<0.01
∆ z-score BMI	4	195	−0.16	0.21	−0.97	0.28	<0.20
∆ Waist circumference	2	379	−0.80	4.22	−34.00	19.00	<0.01
∆ Waist circumference	3	275	−0.07	4.39	−13.50	19.00	<0.05
∆ Waist circumference	4	192	1.14	5.11	−12.00	32.00	<0.01
∆ Hip circumference	2	379	−0.53	3.64	−15.00	16.00	<0.01
∆ Hip circumference	3	275	0.99	4.60	−30.00	22.00	<0.01
∆ Hip circumference	4	192	2.31	4.02	−7.00	14.00	<0.20
∆ WHR	2	379	−0.004	0.062	−0.530	0.210	<0.01
∆ WHR	3	275	−0.011	0.066	−0.290	0.450	<0.01
∆ WHR	4	192	−0.011	0.066	−0.260	0.500	<0.01

$\bar{x}$: average; n: size of the group; SD: standard deviation; min.: minimum value; max.: maximum value; *p*: probability; ∆: the change.

**Table 3 ijerph-19-15899-t003:** Variables of time of consecutive specialized interventions.

Parameter	Visit Number	n	$\bar{x}$	SD	Min.	Max.	*p*
∆ Time	2	394	84.58	23.58	48.99	228.08	<0.01
∆ Time	3	278	194.56	52.35	136.93	725.02	<0.01
∆ Time	4	195	391.64	75.83	277.95	828.97	<0.01

$\bar{x}$: average; n: size of the group; SD: standard deviation; min.: minimum value; max.: maximum value; *p*: probability; ∆: the change.

**Table 4 ijerph-19-15899-t004:** Anthropometric parameters of children in group 2.

Visit Number	n	$\bar{x}$	SD	Min.	Max.	*p*	R
BMI percentile							
1	394	95.33	2.64	90.00	99.90	<0.01	0.18
2	391	94.17	3.55	72.00	99.90		
Z-score BMI							
1	394	1.89	0.30	1.33	2.76	<0.01	0.15
2	391	1.80	0.32	0.94	2.71		

$\bar{x}$: average; n: size of the group; SD: standard deviation; min.: minimum value; max.: maximum value; *p*: probability; R: correlation coefficient.

**Table 5 ijerph-19-15899-t005:** Anthropometric parameters of children in the third group.

Visit Number	n	$\bar{x}$	SD	Min.	Max.	*p*	R
BMI percentile							
1	278	95.34	2.58	90.00	99.90	<0.01	0.20
2	276	94.28	3.19	85.00	99.90		
3	277	93.73	4.02	77.00	99.90		
Z-score BMI							
1	278	1.89	0.30	1.33	2.76	<0.01	0.17
2	276	1.80	0.31	1.09	2.71		
3	277	1.76	0.34	0.92	2.70		
Hip circumference							
1	276	81.28	6.20	69.00	103.00	<0.01	0.09
2	270	80.98	6.08	66.00	104.00		
3	276	82.23	6.36	62.00	108.00		
WHR index							
1	276	0.89	0.06	0.71	1.14	0.15	0.07
2	270	0.88	0.06	0.60	1.08		
3	276	0.88	0.07	0.67	1.45		

$\bar{x}$: average; n: size of the group; SD: standard deviation; min.: minimum value; max.: maximum value; *p*: probability; R: correlation coefficient.

**Table 6 ijerph-19-15899-t006:** Change in time and anthropometric parameters in group 3.

Visit Number	n	$\bar{x}$	SD	Min.	Max.	*p*	R
∆ BMI percentile							
2	276	−1.06	1.87	−10.00	4.00	<0.01	0.11
3	277	−1.59	2.86	−17.00	5.00		
∆ z−score BMI							
2	276	−0.09	0.14	−1.16	0.23	0.01	0.11
3	277	−0.13	0.19	−0.87	0.47		
∆ waist circumference							
2	268	−0.83	4.33	−34.00	19.00	0.04	0.09
3	275	−0.07	4.39	−13.50	19.00		
∆ hip circumference							
2	268	−0.44	3.68	−15.00	14.00	<0.01	0.17
3	275	0.99	4.60	−30.00	22.00		

$\bar{x}$: average; n: size of the group; SD: standard deviation; min.: minimum value; max.: maximum value; *p*: probability; R: correlation coefficient; ∆: the change.

**Table 7 ijerph-19-15899-t007:** Anthropometric parameters of children in group four.

Visit Number	n	$\bar{x}$	SD	Min.	Max.	*p*	R
BMI							
1	195	22.56	2.31	18.70	33.80	0.06	0.10
2	193	22.32	2.38	18.83	34.85		
3	194	22.46	2.42	18.59	35.60		
4	195	22.95	2.58	18.01	37.34		
BMI percentile							
1	195	95.41	2.42	90.00	99.90	<0.01	0.21
2	193	94.45	2.88	85.00	99.90		
3	194	93.91	3.50	82.00	99.90		
4	195	93.37	4.57	74.00	99.90		
Z-score BMI							
1	195	1.89	0.28	1.33	2.74	<0.01	0.19
2	193	1.80	0.29	1.18	2.71		
3	194	1.76	0.32	1.07	2.63		
4	195	1.72	0.35	0.67	2.65		
Waist circumference							
1	194	72.02	6.70	58.00	95.00	0.07	0.10
2	190	71.34	6.47	59.00	95.00		
3	194	71.55	6.74	57.00	98.00		
4	193	73.04	7.14	56.00	101.00		
Hip circumference							
1	194	81.19	6.02	69.00	103.00	<0.01	0.16
2	190	80.77	5.63	69.00	104.00		
3	194	81.99	6.30	62.00	108.00		
4	193	83.42	6.45	69.00	110.00		
WHR							
1	194	0.89	0.07	0.71	1.14	0.09	0.09
2	190	0.88	0.06	0.73	1.04		
3	194	0.87	0.06	0.67	1.26		
4	193	0.88	0.06	0.73	1.28		

$\bar{x}$: average; n: size of the group; SD: standard deviation; min.: minimum value; max.: maximum value; *p*: probability; R: correlation coefficient.

**Table 8 ijerph-19-15899-t008:** Change in anthropometric parameters in group 4.

Visit Number	n	$\bar{x}$	SD	Min.	Max.	*p*	R
∆ Czas							
2	195	81.10	18.29	55.93	150.06	<0.01	0.93
3	195	189.95	42.47	137.04	628.97		
4	195	391.64	75.83	277.95	828.97		
∆ BMI							
2	193	−0.24	0.92	−7.46	1.74	<0.01	0.23
3	194	−0.10	1.11	−5.96	2.80		
4	195	0.39	1.38	−5.58	4.53		
Centyl BMI							
2	193	−0.97	1.77	−10.00	3.00	<0.01	0.16
3	194	−1.49	2.39	−12.00	2.00		
4	195	−2.04	3.57	−20.00	6.00		
Z−score BMI							
2	193	−0.09	0.15	−1.16	0.18	<0.01	0.17
3	194	−0.13	0.17	−0.87	0.24		
4	195	−0.16	0.21	−0.97	0.28		
∆ waist circumference							
2	189	−0.78	3.79	−12.00	12.00	<0.01	0.19
3	193	−0.44	3.87	−13.00	11.00		
4	192	1.14	5.11	−12.00	32.00		
∆ hip circumference							
2	189	−0.50	3.67	−15.00	10.00	<0.01	0.27
3	193	0.86	4.60	−30.00	22.00		
4	192	2.31	4.02	−7.00	14.00		

$\bar{x}$: average; n: size of the group; SD: standard deviation; min.: minimum value; max.: maximum value; *p*: probability; R: correlation coefficient; ∆: the change.

## Data Availability

Data available upon inquiry.

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
