# Peer review of "Effects of 12-Month Interdisciplinary Interventions in 8- and 9-Year-Old Children with Excess Body Weight"

_ijerph, 2022, doi:10.3390/ijerph192315899_

Round 1
Reviewer 1 Report
Dear authors, thanks for submitted your contribution to the IJERPH. Your work is interesting but there are missing information to improve your manuscript. For that reason my recommendation is: Reconsider after major revision (control missing in some experiments.
You can find my comments in the next lines.
- Abstract. Please review it because there are some words with no meaning in English.
- Introduction. This section is very short. I know, there are lot of information about childhood obesity and also, is a well know topic for many scientistic and clinicians, but your main contribution in this paper is about an intervention to reduce bodyweight, for that reason please expand your introduction about the benefits, disadvantages of interventions like yours, also add more data about previous papers with similar intervention.
- Patients and methods. In the study design section please be more concise and organize, is very difficult to follow the process of your research. Also, you duplicate information in this section and in the following one. For example you can organize in the next subheading: study design, sample size and selection criteria; nutritional assessment (anthropometric, blood pressure, body composition analysis and physical assessment); intervention characteristics; statistical analysis. Is not clear why subanalize the data by group or number or visit. You can use some logistic regression to adjust by number of visit for example.
- Results. Please organize this section according to your aims. 1st part about obesity prevalence and second part about the results from the intervention. You have to improve the data presentation to understand the results, for example using some graphs, or a more organized table. In your results is not clear if the intervention is effective or not.
Author Response
Dear authors, thanks for submitted your contribution to the IJERPH. Your work is interesting but there are missing information to improve your manuscript. For that reason my recommendation is: Reconsider after major revision (control missing in some experiments.
You can find my comments in the next lines.
- Abstract. Please review it because there are some words with no meaning in English.
Thank you very much for this observation. The abstract has been corrected.
- Introduction. This section is very short. I know, there are lot of information about childhood obesity and also, is a well know topic for many scientistic and clinicians, but your main contribution in this paper is about an intervention to reduce bodyweight, for that reason please expand your introduction about the benefits, disadvantages of interventions like yours, also add more data about previous papers with similar intervention.
Thank you very much for this remark. We have added the missing topics to the introduction.
- Patients and methods. In the study design section please be more concise and organize, is very difficult to follow the process of your research. Also, you duplicate information in this section and in the following one. For example you can organize in the next subheading: study design, sample size and selection criteria; nutritional assessment (anthropometric, blood pressure, body composition analysis and physical assessment); intervention characteristics; statistical analysis. Is not clear why subanalize the data by group or number or visit. You can use some logistic regression to adjust by number of visit for example.
The information in section „Patients and methods” is ow more organised. Another subchapters have been added to make the text clearer and more accessible to the reader. Additionally, the chapter “Methods” was expanded, as suggested by another reviewer.
- Results. Please organize this section according to your aims. 1st part about obesity prevalence and second part about the results from the intervention. You have to improve the data presentation to understand the results, for example using some graphs, or a more organized table. In your results is not clear if the intervention is effective or not.
We have organised the conclusions according to your suggestions, and corrected the way of presenting the results. Additionally some charts were added, which makes the result clearer and more connected to the conclusions.
Reviewer 2 Report
The Introduction section needs extensive modification. As currently written, authors do not provide enough details to the readers about the problem statement. Authors must conduct additional literature review and elaborate on this problem thus laying a foundation for the design of their study. Authors must include information on the uniqueness (if any) of their study/study design.
Minor English language errors in the manuscript need to be addressed.
Author Response
The Introduction chapter has been revised, expanding the topic in more detail. We have supplemented the information regarding the conduct of intervention programs of a similar nature to ours to bring the reader closer to the leading theme of the work. The uniqueness of our project lies in the fact that the screening stage involved a very large population (4890 children) and each child received an individually prepared report on potential deviations in the tests performed. In addition, the project had a very high social significance and was the first study conducted in our region ( this information has been added to the text). Attempts have been made to correct minor errors resulting from the translation of the text that have appeared in the manuscript.
Reviewer 3 Report
This study identifies the prevalence of overweight and obesity in 8-and 9-year children in Szczecin and examines the effectiveness of a 12-month interdisciplinary on reducing body weight in overweight/obese children. Thank you for the opportunity to review this work, and my comments are:
For any words, full spelling should be provided when the abbreviation first appears. Please check the whole manuscript, including Abstract and tables.
Abstract.
Please delete words in lines 20-21 that are not related to the research contents.
Introduction
Contents of the Introduction are too short to give readers a general understanding of the current research. Also, research gap that tell readers why authors conduct this study was not identified.
Study design
As this study includes a cross-sessional observation and an intervention, the description of the study design therefore should be separated according to different designs.
Projected and real number of children
(1) Detailed inclusion and exclusion criteria to recruit participants should be provided.
(2) What is the age range for the 4890 children attending the “first stage” of the study?
(3) Authors should double check data that used to estimate the prevalence of overweight/obesity. Because, of the 4890 children attended the “first stage”, 86 belonged to overweight. Thus, the prevalence of overweight should be 1.8%. But the authors presented the prevalence was 16.9%. Also, the prevalence of obesity should be 6.2%, but the authors reported the prevalence was 6.4%.
Methods
(1) Detailed intervention contents should be provided.
(2) Some outcomes (e.g., blood pressure, percentage body fat, and physical fitness) that didn’t relate to the research aim were measured. And there were not any results in relation to these outcomes were reported. So authors should consider if it is necessary to keep these outcomes in the manuscript.
Statistical analysis
(1) Please delete the repeated sentences in Line 162-163.
(2) Authors presented “a logistic model was used to estimate what factors influence the increase of variables”. However, the factors and data were identified and reported in Results.
Author Response
This study identifies the prevalence of overweight and obesity in 8-and 9-year children in Szczecin and examines the effectiveness of a 12-month interdisciplinary on reducing body weight in overweight/obese children. Thank you for the opportunity to review this work, and my comments are:
For any words, full spelling should be provided when the abbreviation first appears. Please check the whole manuscript, including Abstract and tables.
Thank you for this observation. We added the full spelling to first appearances of each abbreviation.
Abstract.
Please delete words in lines 20-21 that are not related to the research contents.
The words appeared in text erroneously, they have now been removed.
Introduction
Contents of the Introduction are too short to give readers a general understanding of the current research. Also, research gap that tell readers why authors conduct this study was not identified.
Thank you very much for this comment. The Introduction chapter has been revised, elaborating on the topic in more detail. We have added information regarding the conduct of intervention programs of a similar nature to ours, in order to bring the reader closer to the leading theme of the work, thus emphasizing the importance of conducting preventive activities. In addition, we also added the information on why we decided to conduct such a study.
Study design
As this study includes a cross-sessional observation and an intervention, the description of the study design therefore should be separated according to different designs.
As per reviewer’s suggestion, this section was divided into part 1 – screening, and part 2 – observation.
Projected and real number of children
- Detailed inclusion and exclusion criteria to recruit participants should be provided.
Detailed criteria were added to the text, as per reviewer’s suggestion.
- What is the age range for the 4890 children attending the “first stage” of the study?
The group consisted of children in the age between 6 and 12, however there were only 8 children above the age of 10, and they were not included in the statistical analysis.
- Authors should double check data that used to estimate the prevalence of overweight/obesity. Because, of the 4890 children attended the “first stage”, 86 belonged to overweight. Thus, the prevalence of overweight should be 1.8%. But the authors presented the prevalence was 16.9%. Also, the prevalence of obesity should be 6.2%, but the authors reported the prevalence was 6.4%.
There was an error in the numer, the group involved 826 and not 86 children with overweight, which makes up for 16.9% of the study group. Obesity was diagnosed in 313 children – 6.4%. Thank you for this observation, corrections were made in the text.
Methods
- Detailed intervention contents should be provided.
As per reviewer’s suggestion, the details of the intervention and all specialists’ role were added to the text.
(2) Some outcomes (e.g., blood pressure, percentage body fat, and physical fitness) that didn’t relate to the research aim were measured. And there were not any results in relation to these outcomes were reported. So authors should consider if it is necessary to keep these outcomes in the manuscript.
Evaluation of some parameters, such as blood pressure, body fat percentage, and physical fitness were part of our project. On the other hand, they were not directly related to the objective we present in this article. We decided not to omit the description of individual actions in the Participants and methods section in order to present the course of the intervention phase. Future publications will include the results of further parameters studied. Thank you for this comment.
Statistical analysis
- Please delete the repeated sentences in Line 162-163.
Repetition was deleted.
- Authors presented “a logistic model was used to estimate what factors influence the increase of variables”. However, the factors and data were identified and reported in Results.
As indicated, the subsection entitled Statistical analysis was corrected by dropping the description of methods that were not used in the presented work.
Reviewer 4 Report
Thank you for your interesting manuscript. However, I have some comments for the better manuscript.
Major comments
1. In the aim of objectives of the study, the authors should change the precise and compact wordings such as the words “to obtain epidemiological data on the ….” and “medical intervention in the form of a year of ….”.
2. The study design and participant selection should be presented as the algorithm.
3. The intervention program was one-year program. Why did you choose the participants for three consecutive years?
4. In page 2 at line 48, you reported that the participants for 2017 were the combination of the participants from 2016 & 2017. As the intervention program was one year, why did you combine the number of participants for 2017 (2016+2017)?
5. In line 87 at page 2, you stated as “to mainly evaluate carbohydrate and lipid metabolism for four visit”. However, you did not mention anything regarding those metabolisms in the results.
6. The expected number of participants were 11,494, you stated. Could you add the reference to get this number of participants in terms of calculation please?
7. The eligible participants were 515. However, I did not clear that these 515 participants were the combination of 3 years?
8. If so, why did you choose the participants for 3 years although the intervention program was one year? The different participants from different years could not be combined for the same analysis generally.
9. Regarding the measurement of arterial pressure, you stated that the measurement was taken after several minutes of rest in ………………….. Actually, you should have the exact rules for timing of blood pressure e.g referencing the WHO STEPS survey guidelines.
10. The logistic regression was stated in your analysis. However, you did not report the results regarding the logistic regression in the results.
11. Which study year of the participants had included in the results showing in the tables?
12. You stated that “the group 2” in the table 4. But, you did not mention any categorization about the group.
Minor comments
1. The title for No.3 as patients and methods are described. Why did you mention the participants as the patients? This title should be under the title of methods.
2. In line 61, page 2, you stated that BMI was included in your measurements. Actually, BMI cannot be measured, can be analyzed only.
3. The reference should be cited as the relevant.
Author Response
Thank you for your interesting manuscript. However, I have some comments for the better manuscript.
Major comments
- In the aim of objectives of the study, the authors should change the precise and compact wordings such as the words “to obtain epidemiological data on the ….” and “medical intervention in the form of a year of ….”.
The wording of the program objectives has been changed to be more precise. Thank you for this comment
- The study design and participant selection should be presented as the algorithm.
The paper now includes a diagram showing how the children were qualified to the study group, which was intended to make it easier for the reader to understand. Study design was divided into subsections to make it more readable as well. Thank you for your comment.
- The intervention program was one-year program. Why did you choose the participants for three consecutive years?
The project was a program of great social importance and conducted on a large scale. It was financed by our city's own funds and was carried out for 3 consecutive years. Due to this fact, we surveyed consecutive year groups of 8- and 9-year-old children for three years. In the study group, there happened to be individual children who were older or younger attending a particular grade, but for ethical and social reasons we could not deny them participation in the program. The medical data of children from the three years of the intervention significantly increased the size of the study group. An explanation of the duration of the project-three consecutive years-is included in the paper in the Projected and real number of children section.
- In page 2 at line 48, you reported that the participants for 2017 were the combination of the participants from 2016 & 2017. As the intervention program was one year, why did you combine the number of participants for 2017 (2016+2017)?
Because the school had a combined 2016+2017 yearbook of children. We did not want to examine only a portion of the class so that no child would feel neglected or singled out. We refer here to the social nature of the program.
- In line 87 at page 2, you stated as “to mainly evaluate carbohydrate and lipid metabolism for four visit”. However, you did not mention anything regarding those metabolisms in the results.
Thank you for this remark, this part was deleted.
- The expected number of participants were 11,494, you stated. Could you add the reference to get this number of participants in terms of calculation please?
These were data from statistical yearbooks. The source is included in the paper.
- The eligible participants were 515. However, I did not clear that these 515 participants were the combination of 3 years?
The number of 515 participants is the total of children who qualified for the second stage of the program during the 3 years of the project. I have added this information to the presented text so that readers will have no doubts. Thank you for this comment.
- If so, why did you choose the participants for 3 years although the intervention program was one year? The different participants from different years could not be combined for the same analysis generally.
The project was a study of great social importance. It was financed by our city's own funds and carried out for 3 consecutive years. In each of these three years, a screening and intervention stage was conducted for a specific year group, such that the main audience was 8 and 9 year old children. Thank you for this comment.
- Regarding the measurement of arterial pressure, you stated that the measurement was taken after several minutes of rest in ………………….. Actually, you should have the exact rules for timing of blood pressure e.g referencing the WHO STEPS survey guidelines.
Corrected as per reviewer’s suggestion.
- The logistic regression was stated in your analysis. However, you did not report the results regarding the logistic regression in the results.
In the end the logistic regression model was not used. Information about its use has been removed from the subsection Statistical analysis methods. The results extended to include this issue and the results of other parameters studied during this project will be presented in the next publication. Thank you for your comment.
- Which study year of the participants had included in the results showing in the tables?
The table includes all the children collected during the 3 years of the program, the total number is 515 children. Thank you for this comment.
- You stated that “the group 2” in the table 4. But, you did not mention any categorization about the group.
The description of the groups in the Study Design section (stage 2) was expanded. Group 1 consisted of children who appeared at only 1 visit, group 2 consisted of children who appeared at the first two visits, group 3 were children , who attended 3 specialty visits, and group 4 were children who completed the program, i.e., attended all 4 visits. . Missing information in the Participants and Methods section has been added.
Minor comments
- The title for No.3 as patients and methods are described. Why did you mention the participants as the patients? This title should be under the title of methods.
- The participants were mistakenly called patients. This misplaced wording was changed throughout the work. The names of chapters and subsections were corrected to make the work more readable.
- In line 61, page 2, you stated that BMI was included in your measurements. Actually, BMI cannot be measured, can be analyzed only.
As suggested, changed to BMI analysis instead of measurements of BMI. Thank you for your comment.
- The reference should be cited as the relevant.
Corrected as per reviewer’s suggestions.
Round 2
Reviewer 1 Report
Dear authors,
Thanks a lot for improve your manuscript. Now my recommendation is accept it in the current form.
Reviewer 2 Report
Authors addressed my comments satisfactorily.
Reviewer 3 Report
All issues that I raised last round have been addressed by authors.